# How to sample the world for understanding the visual system

**Johannes Roth (jroth@cbs.mpg.de)**, **Martin N. Hebart (hebart@cbs.mpg.de)**
Max Planck Institute for Human Cognitive and Brain Sciences, 04103 Leipzig, Germany
Department of Medicine, Justus Liebig University Giessen, 35390 Giessen, Germany
Center for Mind, Brain and Behavior (CMBB), Universities of Marburg, Giessen, and Darmstadt

## Abstract

**Understanding vision requires capturing the vast diversity of the visual world we experience. How can we sample this diversity in a manner that supports robust, generalizable inferences? While widely-used, massive neuroimaging datasets have strongly contributed to our understanding of brain function, their ability to comprehensively capture the diversity of visual and semantic experiences has remained largely untested. More broadly, the factors required for diverse and generalizable datasets have remained unknown. To address these gaps, we introduce LAION-natural, a curated subset of 120 million natural photographs filtered from LAION-2B, and use it as a proxy of the breadth of our visual experience in assessing visual-semantic coverage. Our analysis of CLIP embeddings of these images suggests significant representational gaps in existing datasets, demonstrating that they cover only a restricted subset of the space spanned by LAION-natural. Simulations and analyses of functional MRI data further demonstrate that these gaps are associated with impaired out-of-distribution generalization. Importantly, our results reveal that even moderately sized stimulus sets can achieve strong generalization if they are sampled from a diverse stimulus pool, and that this diversity is more important than the specific sampling strategy employed. These findings not only highlight limitations of existing datasets in generalizability and model comparison, but also provide guidance for future studies to support the development of stronger computational models of the visual system and generalizable inferences.**

**Keywords:** stimulus selection; naturalistic fMRI; adaptive sampling; dataset; generalization

## Introduction

Humans encounter an incredibly diverse range of visual stimuli, and capturing this breadth is essential for understanding how the brain represents visual information. This has motivated the development of ever larger datasets of brain activity in response to naturalistic visual stimuli (Allen et al. (2022); Chang et al. (2019); Hebart et al. (2023)). While broad sampling suffers from diminishing returns (Allen et al. (2022)), large-scale, condition-rich datasets have laid the foundation for extensive computational modeling of the visual system, allowing for a detailed and fine-grained analysis of its function (Contier et al. (2024); Takagi & Nishimoto (2023)).

While these datasets have been invaluable in advancing our understanding, particularly within specific domains like object and scene processing, their ability to capture the visual-semantic diversity of our world remains largely untested. This has an important consequence: If a dataset lacks diversity, then inferences may not generalize. This is particularly relevant in cognitive computational neuroscience, where recent insights from model comparison, encoding, and decoding often rely on such datasets (Doerig et al. (2023)). While previous research has highlighted a lack of semantic diversity in some datasets (Shirakawa et al. (2024)), much less is known about how the visual-semantic coverage of real-world concepts affects generalization performance.

To address these gaps, we used a three-pronged approach. First, we assessed how well existing stimulus sets cover the breadth of visual experience by embedding them within a much larger representational image space defined by LAION-natural, a newly curated subset of 120 million naturalistic photographs (images depicting real-world scenes or objects, excluding heavily edited or synthetic content) filtered from LAION-2B (Schuhmann et al. (2022)). Second, we evaluated whether a sufficiently diverse stimulus set can enable generalization at a scale practical for vision research. Finally, in simulations and analyses of functional MRI data, we determined the effect of sampling strategy on generalization performance.

## Methods

### Constructing and evaluating LAION-natural

**Filtering LAION-2B to naturalistic images** LAION-2B (Schuhmann et al. (2022)) contains ~2 billion images with English captions, but visual inspection revealed a large fraction of unsuitable, non-natural images. We filtered images using metadata, removing NSFW images (often problematic for general participant studies) and those with any dimension smaller than 400px, leaving ~720 million images. Of these, ~440 million were still accessible via URL.

We next removed non-naturalistic images, by establishing three exclusion criteria: watermarks or banners, heavy editing (e.g., strong image filters), and not a real-world scene or object. Based on these, we manually labeled 25,000 images from a pool of 200,000 random LAION-2B samples. Labeling was initialized by first clustering the pool into 400 clusters with mini-batch k-Means on CLIP features from OpenAI's CLIP ViT-32B. These were then manually split into "natural", "not natural" and "mixed", from which 5,000 images were selected from "natural" and "non-natural" clusters. Using entropy-based uncertainty sampling, we iteratively identified the most informative samples to label. Labeling was stopped when accuracy plateaued (5-fold cross-validated). From the resulting dataset, we trained a logistic regression classifier on CLIP features (natural/non-natural), achieving an accuracy of ~82%. For

filtering, we used a higher probability threshold to achieve a precision of 90%. Removing images that were labeled non-naturalistic by the classifier left us with ~120 million images to use for evaluating existing datasets and simulations.

Analyzing random subsets of LAION-natural (6k samples, 10 seeds, as in Fig. 1D), we found semantic coverage of 93.14% of LAION-2B (filtered only for NSFW and resolution, see "Evaluating coverage"), suggesting retained visual-semantic diversity. This was further confirmed by measuring the OOD performance of random NSD-sized subsets of LAION-2B (as in Fig. 2E), which outperformed LAION-natural by only 2.34%.

**Ensuring semantic richness of LAION-natural**  To validate that LAION-natural contains all concepts found in ImageNet (Deng et al. (2009)), MS COCO (Lin et al. (2014)), and ecoset (Mehrer et al. (2021)), we extracted text features for each concept and built an approximate nearest-neighbor search index on 5 million randomly sampled LAION-natural images. For each concept, we retrieved the 100 most similar images, based on the cosine similarity of normalized image and text embeddings. Manual inspection confirmed that every concept had at least one corresponding image in LAION-natural.

## Evaluating coverage

We clustered 10 million random LAION-natural CLIP samples into 5,500 clusters using mini-batch k-Means. Cluster centroids were projected into 2D using t-Distributed Stochastic Neighbor Embedding, t-SNE (Van Der Maaten & Hinton (2008)). We further quantified coverage by looking at the opposite metric, namely measuring how many images in LAION-natural would be considered outliers with respect to the feature space of established datasets like THINGS or NSD. To this end, we used PCA on the CLIP features of the images. PCA was first trained on the features of the "covering" dataset (e.g., THINGS or NSD), with the number of principal components selected to retain 95% of the variance in that dataset. This step effectively created a lower-dimensional representation of the dataset's feature distribution.

Next, we calculated the PCA reconstruction error for each image in both the "covering" dataset and the "covered" dataset (LAION-natural). This error quantified how well an image can be reconstructed from the principal components learned from the "covering" dataset. A higher error suggests the image's features are not well captured by that PCA space. To establish a criterion for whether a LAION-natural image was covered by the THINGS/NSD feature space, we defined an error threshold by fitting a generalized extreme value (GEV) distribution to the reconstruction errors of the "covering" dataset itself and selecting the 95th percentile from this fitted distribution. A LAION-natural image was considered covered if its reconstruction error, when projected into and reconstructed from the "covering" dataset's PCA space, was below this error threshold. The final coverage percentage was reported as the proportion of "covered" LAION-natural images relative to the total number of tested LAION-natural images.

## Simulating OOD generalizability

**GMM-based simulation**  To evaluate how OOD accuracy changed with dataset diversity and size, we simulated stimulus features and brain responses using a Gaussian mixture model (GMM) with 100 clusters in a high-dimensional space ($D = 512$, same as dimensionality of CLIP embeddings). Cluster centers $\mu_c$ were drawn from $N(0, \sigma_{inter}^2 * I)$, with $\sigma_{inter}^2 = 100/D$, and samples were generated from $N(\mu_c, \sigma_{intra}^2 * I)$, with $\sigma_{intra}^2 = 10/D$, ensuring well-separated and evenly spaced clusters. From these synthetic stimulus features $y$, brain response $x$ was generated via a linear mapping, a common assumption for both encoding and decoding models of brain activity. We specifically used a random teacher weight matrix $A$ ($N(0, 1/\sqrt{D})$), adding Gaussian noise $\varepsilon$ with variance $\sigma_{noise}^2 = 0.25$:

$$x = A^T * y + \varepsilon$$

To assess generalization, we generated 100 samples from a new cluster and measured how often the predictions aligned with the centroid of the OOD cluster. This was repeated 32 times with different OOD clusters for robust estimates.

**Simulation from CLIP feature space**  We extended the previous analysis to image features from LAION-natural, keeping brain response generation unchanged. We ran mini-batch k-Means clustering (k = 1,000) on CLIP features derived from LAION-natural and then used agglomerative clustering to group them into six coherent cluster groups. These groups served to divide the feature space into in-distribution and out-of-distribution folds. We measured OOD accuracy by training a model on 6,000 samples from varying parts of the in-distribution space (from combinations of 2, 3, 4, or 5 of the cluster groups, repeated 100 times per condition) and tested on 1,000 OOD samples from the OOD group, by measuring how often predictions aligned with the centroid of their source cluster (cross-validated across all cluster groups).

## Sampling strategies

**(Stratified) random sampling**  Samples were drawn uniformly at random from the entire pool of stimuli, ensuring each selection was unique (sampling without replacement). Alternatively, for stratified random sampling, the aim was to achieve proportional representation from predefined stimulus clusters. An approximately equal number of samples (n_samples//number of clusters) was drawn from each cluster. If a cluster contained fewer samples, all available samples from that cluster were selected. Both random and stratified sampling were repeated 100 times per dataset size for robust performance estimates.

**k-Means-based sampling**  The stimulus pool was clustered using mini-batch k-Means, with the number of clusters set to be the number of stimuli to sample. Data points closest to each resulting cluster centroid (in terms of Euclidean distance) were selected. To increase the speed of finding these nearest neighbors for each centroid, an Annoy index was built using

the stimulus features (n_trees $= 50$).

**Core-Set sampling**   We iteratively selected samples to minimize the maximum distance between any data point and its nearest selected point (Sener & Savarese (2018)), broadly covering the available feature space of the dataset. We used the kCenterGreedy algorithm. This greedy algorithm starts from an empty set and selects the first sample randomly. Subsequently, it iteratively adds the data point from the remaining pool that is farthest from any of the points already selected into the core-set. This ensures that each newly added sample maximally reduces the coverage radius of the selected set, and therefore increases diversity and representation of the overall feature space.

**Sampling to optimize effective dimensionality**   Effective dimensionality (ED) measures the number of meaningful axes of variance in a dataset (Del Giudice (2021)). We used the participation ratio of CLIP image features to estimate ED:

$$\text{ED} = \frac{(\sum_{i=1}^{K} \lambda_i)^2}{\sum_{i=1}^{K} \lambda_i^2}$$

where $\lambda_i$ are the principal components. Intuitively, a low ED suggests an over-representation of semantic concepts - for example, if a dataset contains only mountains and beaches, variance is mostly explained by a single "beach-or-mountain" dimension. A more diverse dataset, also containing meadows, forests, or cities, would require more dimensions.

Given this insight, we also greedily sampled to maximize the ED of the dataset. Initialization of the selected set was performed in one of two ways: either by selecting two samples uniformly at random, or by using samples closest to cluster centroids derived from a mini-batch k-Means clustering. Samples were added iteratively. In each step, a candidate pool was generated by drawing 10 random samples from each cluster of a precomputed clustering. To avoid selecting very similar items, candidates that were too close (Euclidean distance $< 0.1$) to already selected samples were filtered out. The ED was then estimated for each remaining candidate, if it were added to the existing image set. This ED calculation was performed with an incremental update formula for the covariance matrix for efficiency and was parallelized across 32 CPU cores to speed up selection. The candidate that yielded the highest ED for the augmented set was then added to the selected samples.

**Margin-based, adaptive sampling**   We also tested if epistemic uncertainty could guide sample selection using a margin-based active learning strategy (Balcan et al. (2007)) with a logistic regression model predicting discretized brain responses from stimulus features. Before model training, the stimulus features (X) were standardized to have zero mean and unit variance.

Sampling was initialized with 100 random samples. Target brain data (Y) was discretized into three equally populated bins per dimension using quantile-based binning (e.g., low, medium, high response categories). A logistic regression model was trained for each target dimension (i.e., for each voxel or ROI whose response was being predicted). A candidate pool was drawn by randomly selecting 10 samples per cluster from a precomputed mini-batch k-Means clustering (k=1,000). For each candidate, bin probabilities were predicted by the trained logistic regression model(s), and uncertainty was measured as the margin between the top two probabilities (a lower margin indicates higher uncertainty, as the model is less decisive). Margins were calculated for each dimension independently for a given candidate, and these margins were then averaged to get a single uncertainty score for that candidate. In each iteration, the 100 images with the lowest average margins (highest uncertainty) were selected and added to the training set, and the model was retrained. This process was repeated until reaching the required dataset size.

## Evaluation of concept distribution

To test to what extent a sample dataset preserved the distribution of concepts of the stimulus pool, we evaluated how the sampling strategies changed the concept distribution using a subset of LAION-natural (100,000 images). As LAION-2B only provides image captions, and no image-level keywords, we first used Gemini 1.5 Flash (8B), configured with generation parameters: temperature=1, top_p=0.95, top_k=40, and max_output_tokens=8192, to list keywords for each image, using the prompt "Describe these images in as many keywords as you like. Return as a list of keywords.". This generated 75,535 unique terms, at a cost of $\sim\$2.54$. We then filtered these keywords to only include concrete nouns (Concreteness $> 4$; Brysbaert et al. (2014)) and availability of natural language frequency (Brysbaert & New (2009)). After filtering, 3,563 keywords remained, which were clustered (HDBSCAN, min_samples=1) into 231 groups, allowing comparison of cluster occurrence depending on sampling strategy.

## Implementation details

CLIP features were extracted using the CLIP ViT-32/B model, provided by OpenAI (https://github.com/openai/CLIP). Approximate nearest-neighbor search was implemented with Annoy (https://github.com/spotify/annoy; n_trees=100). For kCenterGreedy, we used Google's active learning framework. Clustering, t-SNE projection, PCA, Ridge, and logistic classifier fitting were implemented using scikit-learn (Pedregosa et al. (2011)). Active-learning classifier training was implemented with AliPy (Tang et al. (2019)).

## Results

### Significant gaps in visual-semantic coverage in existing stimulus sets

Determining how well existing stimulus sets represent our natural visual experience is not possible due to the lack of a detailed understanding of its contents and the relative frequency of different concepts or "classes" of experience. To approximate broader coverage, we used the large and highly diverse LAION-2B dataset ($\sim$2 billion image-text pairs). To focus on

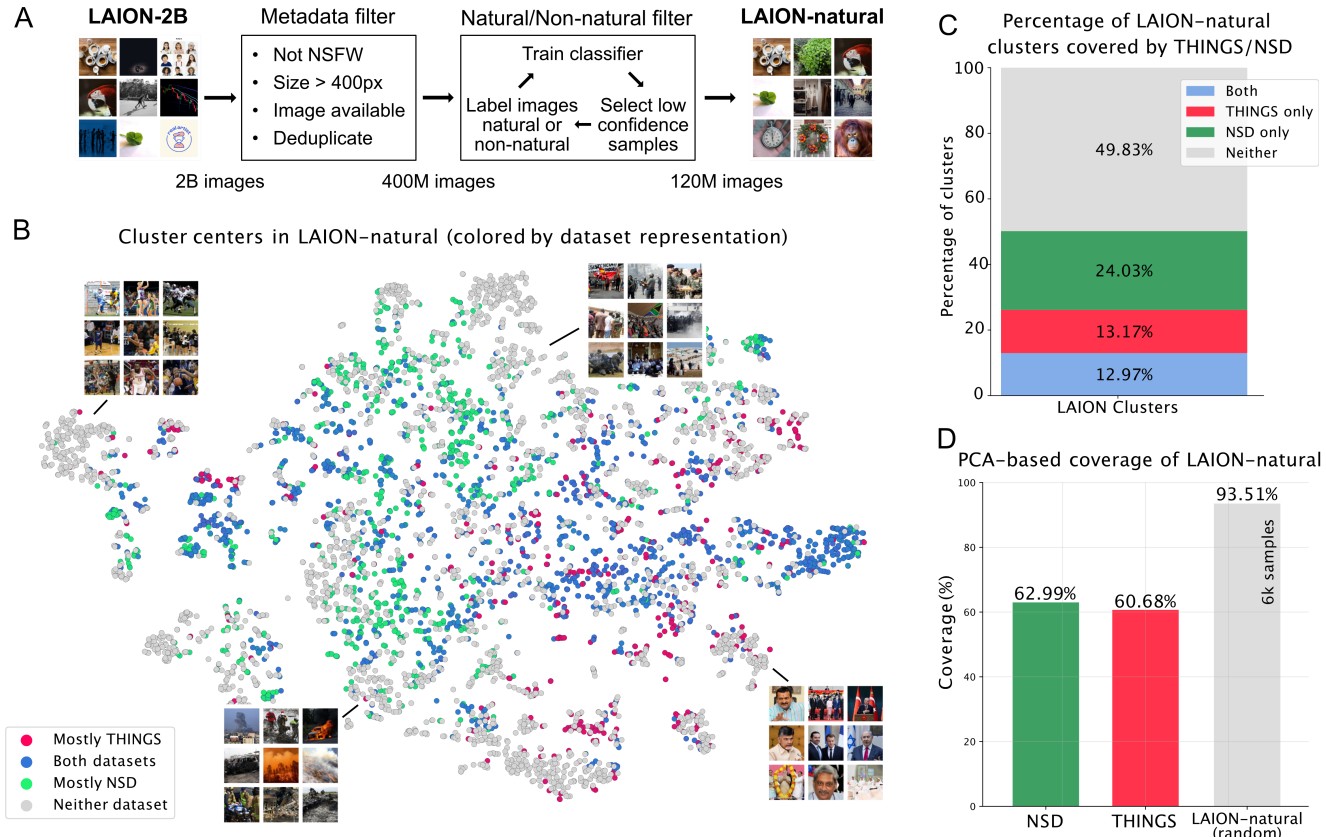

Figure 1: A) Filtering procedure to generate LAION-natural from LAION-2B. B) t-SNE projection of 5,500 cluster centroids from LAION-natural, with colors reflecting presence of existing datasets (cutoff: 2 or more images). Inspection of these clusters revealed various concepts not covered by THINGS or NSD, including landscapes, natural events, crowds, or non-Western public figures (see Fig. S3). C) Percentage of clusters covered by THINGS, NSD, or both (cutoff: 2 or more images). D) Percentage of LAION-natural that is in-distribution, based on Principal Component Analysis (PCA)-based outlier detection. While neither THINGS nor NSD was able to explain the variance in LAION-natural, a random subset of only 6,000 LAION-natural samples still captured 93.51%.

high-quality natural photographs, which more closely reflect our visual experience, we (1) filtered LAION-2B to select only unique, high-resolution images and excluded NSFW content and (2) applied a classifier trained on 25,000 hand-labeled images to restrict images to natural photographs (Fig. 1A, see Fig. S1 for an overview of natural/non-natural images). We term this new image set "LAION-natural" (~120 million photographs). To verify the diversity of LAION-natural, we tested if all concepts found in common stimulus sets, including ImageNet (Deng et al. (2009)), MS COCO (Lin et al. (2014)), and ecoset (Mehrer et al. (2021)), could be found in the dataset. Even though this approximation of natural vision is likely an incomplete characterization and will contain specific biases inherent in the dataset, we can treat coverage and generalization abilities on LAION-natural as a proxy for coverage of the visual world to understand limitations in existing datasets and reveal strategies for broader stimulus sampling.

Having curated a large pool of natural photographs, we next evaluated how much of LAION-natural is covered by the Natural Scenes Dataset (NSD; Allen et al. (2022)) and THINGS (Hebart et al. (2023)), two of the largest, densely sampled visual neuroimaging datasets. We approximated visual-semantic coverage using CLIP-extracted image features, known for their large-scale training datasets and their alignment with both human perceptual ratings (Demircan et al. (2023); Kaniuth et al. (2024); Muttenthaler et al. (2022)) and cortical activity patterns (Conwell et al. (2024); Wang et al. (2023)). To evenly distribute the dataset into similarly sized chunks, we divided LAION-natural into 5,500 clusters using mini-batch k-Means on the image features.

A 2D t-SNE projection of the cluster centroids revealed low overlap between THINGS and NSD (Fig. 1B), likely due to their distinct focuses on scenes and objects, respectively. The slightly higher coverage of NSD can in part be explained by its larger dataset size (70,000 vs. 26,107 images). More importantly, the visualization indicates that both datasets exhibited

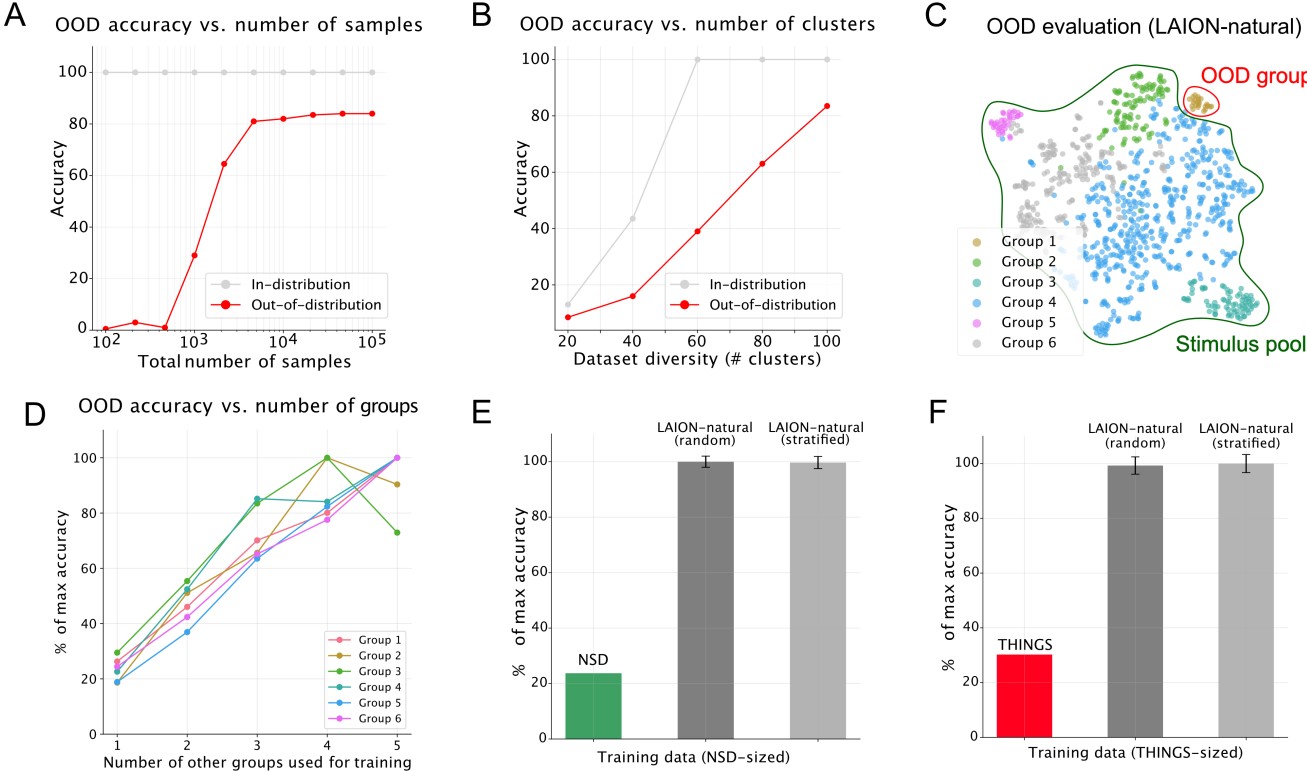

Figure 2: A) OOD accuracy depending on the number of training samples (distributed across 100 clusters). OOD accuracy saturates at 10,000 samples. B) Reducing the number of clusters, while keeping the number of samples constant (6k), reduced OOD accuracy. C) Real-world test for OOD accuracy. LAION-natural is divided into six cluster groups. One of the cluster groups is considered OOD and the remaining groups are used to construct the training set. D) Accuracy for the OOD group from LAION-natural, based on the number of clusters in the training set. Less diverse training sets result in lower OOD accuracy. E & F) OOD accuracy of THINGS and NSD, compared against random and stratified samples from LAION-natural (averaged across groups, controlled for dataset size). Neither of the datasets sufficiently spans the LAION-natural image space to enable OOD generalization.

significant gaps in the visual-semantic image space defined by LAION-natural. We quantified this observation by determining the percentage of LAION-natural clusters represented by at least 2 images from either THINGS or NSD, which ensures an inclusive threshold while minimizing the effect of outliers (Fig. 1C). This analysis showed that 49.83% of LAION-natural clusters were not covered by either dataset. To test whether this finding arises through the direction of comparison, we also clustered both NSD and THINGS (200 clusters, average cluster size 0.5% of total dataset) and assessed how many of them were covered by 10 million random images from LAION-natural. We found that 97.5% of NSD and 100% of THINGS clusters were assigned at least one LAION-natural image, with the few uncovered NSD clusters containing only single images. We additionally evaluated how many of the images in LAION-natural would be considered "in-distribution" for THINGS and NSD, respectively, using a PCA-based reconstruction error approach. This revealed that only 62.99% and 60.68% of LAION-natural were in-distribution for THINGS and NSD, respectively (Fig. 1D).

In contrast, a small random subset of LAION-natural (6,000 samples) already achieved an in-distribution score of 93.51%. Together, these findings demonstrate notable limitations in the semantic diversity and coverage of existing stimulus sets.

## When diversified, even moderate-sized stimulus sets can generalize well

Given the limited visual-semantic diversity in existing large-scale fMRI datasets, we next asked how this affects generalizability of inferences drawn using these data and to what degree it is possible to collect diverse data at realistic scales for future studies. Prior work has shown that a lack of diversity can hinder our ability to draw generalizable conclusions from one of these fMRI datasets (Shirakawa et al. (2024)). However, it is unclear whether these challenges can even be mitigated with realistic dataset sizes. While simulations suggest that covering key representational axes can support out-of-distribution (OOD) generalization for brain-to-image reconstruction (Shirakawa et al. (2024)), these findings were based on 500,000 samples, an impractical scale for most fMRI stud-

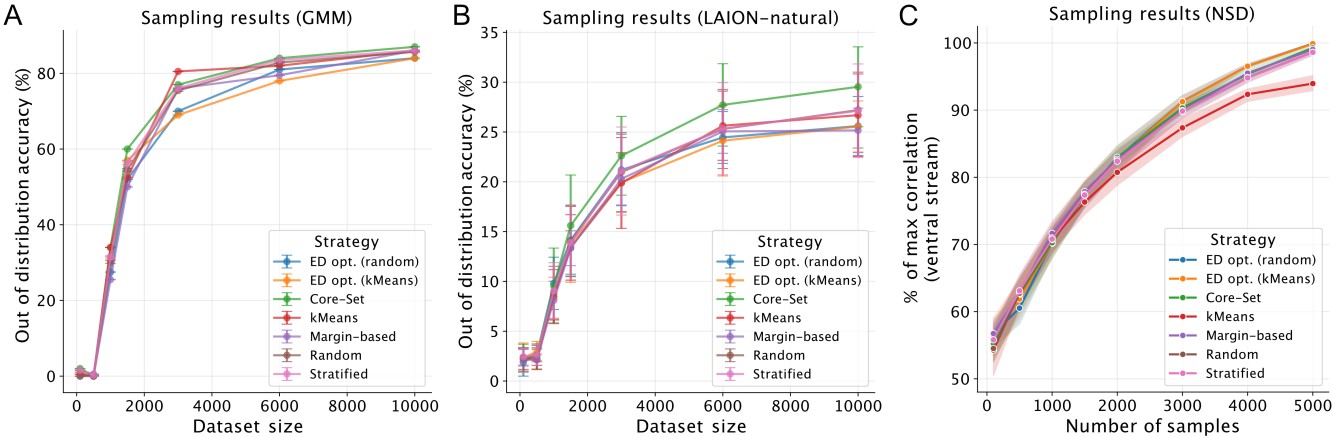

Figure 3: A) From a pool of 500,000 synthetic samples (GMM, C=100), up to 10,000 samples were selected using different sampling strategies (see Methods for details) and evaluated on OOD performance. B) Samples drawn from LAION-natural and evaluated on a held-out cluster group (averaged across groups). C) Effect of sampling strategy on encoding performance in subsets of NSD. Performance is 5-fold cross-validated (20% of NSD as test set), normalized within subjects and averaged across subjects.

ies. Thus, the critical questions of dataset size and dataset diversity on drawing generalizable inferences from fMRI data have remained largely unanswered.

To address these questions, we simulated synthetic fMRI responses to determine how much data is required for OOD generalization. If a lot of data is required, this indicates that it is not feasible in practice to achieve OOD generalization, while if generalization is possible with fewer samples, this highlights the potential of diverse sampling. We used a teacher-student learning model, inspired by previous simulations (Shirakawa et al. (2024)), and generated stimulus features from 100 clusters via a Gaussian mixture model (GMM). Next, we mapped them to synthetic brain responses using a fixed teacher weight plus Gaussian noise. Finally, we employed Ridge regression for decoding features from synthetic brain data. We evaluated predictions on novel OOD clusters using cluster accuracy, i.e., how often predicted features correlated most with the source cluster centroid.

Adjusting the total number of generated samples showed that accuracy started to saturate after 5,000-10,000 samples (distributed across all 100 clusters, Fig. 2A). To determine the effect of dataset diversity for a dataset with realistic size, we fixed the number of samples to 6,000 and varied the number of clusters from which we sampled. This analysis revealed that reducing training set diversity also reduced generalizability (Fig. 2B). Together, these findings show that, in principle, it is possible to identify medium-sized, diverse stimulus sets that can generalize well in neuroimaging studies, as long as the underlying stimulus pool is diverse.

However, the simulations assumed evenly distributed clusters, which does not reflect distributions in real-world datasets. To incorporate dataset realism, we modified our approach by replacing GMM-generated samples with CLIP features from

LAION-natural to generate synthetic brain responses. To allow for broad yet homogeneous sampling across the entire dataset, we clustered images into 1,000 clusters. To simulate the effect of uneven distribution and OOD generalization, we formed 6 OOD groups from these clusters by applying a second layer of clustering. We then used held-out clusters from LAION-natural to evaluate OOD accuracy (Fig. 2C), training the regression model on 6,000 subsamples of the remaining 5 groups. Importantly, to test for uneven sampling, we varied the number of groups sampled from between 1 and 5. The results of this simulation are shown in Fig. 2D, revealing a high accuracy given the noise in the data, but only when incorporating broad sampling across most groups. The results confirm that, to achieve the highest generalization performance, high dataset diversity is required.

While our previous simulations examined the effects of non-diverse datasets with synthetic subgroups, they did not reflect real, empirical datasets. Thus, we extended this analysis by using THINGS and NSD as training datasets, excluding samples assigned to the OOD group that served as a test set. As a baseline, we sampled from LAION-natural using both random and stratified approaches across clusters. The results revealed that THINGS and NSD strongly underperformed in OOD accuracy compared to stratified sampling across all training clusters in LAION-natural, and also, perhaps surprisingly, when sampled randomly (Fig. 2E/F). Additionally, we assessed how dataset diversity affects OOD performance in real fMRI data, by repeating the clustering analysis on NSD, which mirrored the results from LAION-natural (Fig. S2).

Together, these findings highlight that gaps in visual-semantic coverage reduce OOD accuracy, both in simulations and in common existing datasets. Importantly, assum-

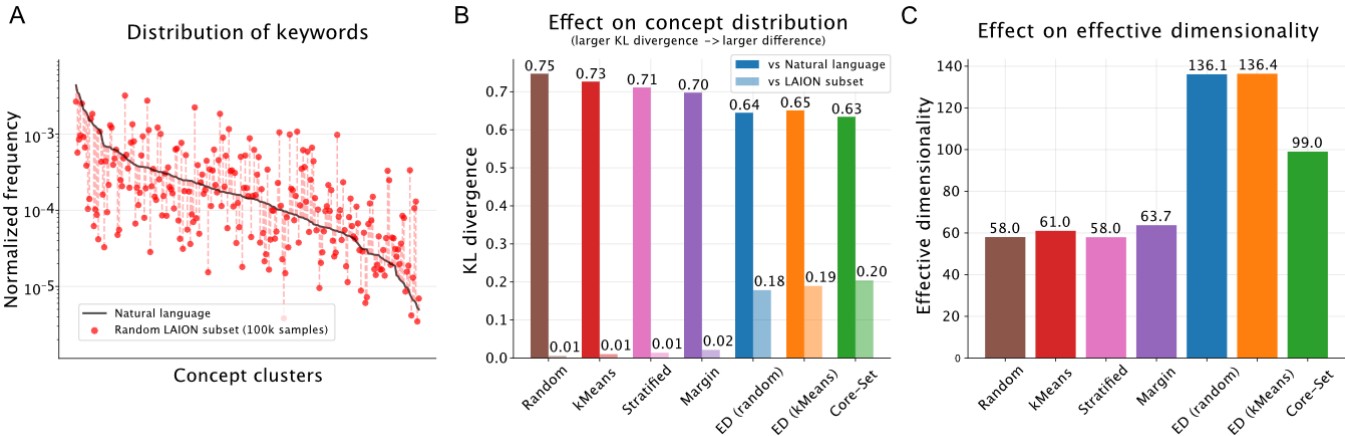

Figure 4: A) Concept distribution of LAION-natural subset (red) differs from natural language frequency (black). B) Depending on sampling strategy, concept distribution of subset is kept (random, stratified, k-Means, margin) or shifts to natural language (Greedy ED, Core-Set). C) Dimensionality of sampled dataset is highest for Greedy-ED, followed by Core-Set and others.

ing results from CLIP embeddings generalize to real fMRI data, our results suggest that constructing a diverse, generalizable stimulus set is feasible within the size constraints of neuroimaging studies.

## Sampling strategy matters less than stimulus pool diversity

Our previous simulation showed that random sampling performed comparably to stratified sampling. This result was unexpected, as targeted sampling could be seen as having a lot of potential for improvements in sampling efficiency and generalizability. Therefore, we addressed the degree to which sampling strategies help to improve dataset efficiency in the presence of a diverse stimulus pool.

To assess the role of sampling strategy in dataset efficiency, we evaluated various established sampling procedures, including random and stratified sampling across clusters, k-Means clustering-based sampling, Core-Set sampling (k-Center-Greedy), greedy sampling to maximize effective dimensionality (ED), and margin-based uncertainty sampling. Similar to our initial simulation, we generated 500,000 samples with a GMM to evaluate the effect of sampling strategy on OOD accuracy across dataset sizes. Our results revealed only minor differences between strategies, with strong effects of dataset size, yet small effects of sampling strategy. Only Core-Set and k-Means-based sampling approaches showed slight advantages at 2,000 and 3,000 data points (Fig. 3A).

To verify these findings with a real image pool, we repeated the sampling experiment using LAION-natural. Unlike previous simulations, where entire groups were left out, we left out sets of clusters, thus approximating sampling from a broad dataset. We found that most sampling strategies performed similarly, except for Core-Set sampling, which consistently outperformed others (Fig. 3B). However, dataset diversity remained the most critical factor, outweighing the choice of sam-

pling strategy.

To test the degree to which these results would generalize to empirical fMRI data, we applied these sampling strategies to NSD and evaluated generalization performance. Rather than focusing on decoding, as in the previous analyses, we focused on encoding, a commonly used approach for NSD. We trained a Ridge regression model to predict single-trial response estimates in the ventral stream using CLIP features (Fig. 3C). Consistent with the previous results, dataset diversity remained the key determinant of performance, while sampling strategy had a minimal impact.

## Sampling strategy can shift concept distribution

The previous results reveal the effect of sampling strategy on generalizability within the stimulus pool, in our case LAION-natural. Being derived from the internet, LAION-natural may have biases in concept frequency distributions. While having only minor effects on generalizability, different sampling strategies could affect if and how these biases translate to sampled datasets. To examine this, we used an LLM to generate word labels for a random subset of LAION-natural (100,000 images) and compared the word frequency distribution to that found in natural language use.

We found that LAION-natural overall does not align well with natural language (see Fig. 4A). By calculating the Kullback-Leibler (KL) divergence between the concept distribution of the sampled dataset and LAION-natural or natural language, we found that random, stratified, k-Means-, and margin-based sampling closely mirrored the stimulus pool distribution, whereas Core-Set and ED-based methods were closer to natural language frequencies (see Fig. 4B). Core-Set and ED-based methods also resulted in the highest increase in the effective dimensionality of resulting image sets, measured via their CLIP embeddings (see Fig. 4C). These results underscore the effectiveness of Core-Set sampling for generating

new datasets. However, the added benefits in generalization remain minor relative to the role of stimulus diversity.

## Discussion

Understanding visual representations requires us to capture much of the visual-semantic richness of the visual world. A prominent research paradigm involves collecting extensive data in response to a broad set of stimuli, with the aim of allowing for generalizable inferences (Naselaris et al. (2021)). Our findings demonstrate that, at the level of visual semantics, commonly used stimulus sets fall short of this goal. Their emphasis on a constrained subset of concepts limits the generalizability of insights that can be drawn from them. However, this does not diminish the value of these datasets, which have significantly advanced our understanding of brain function, particularly within the domains of scenes or objects. And while our study focused on visual semantics, many studies using these datasets were not carried out at that level, and it is possible that existing datasets already have sufficient diversity to comprehensively capture purely low-level and mid-level processing. Future studies should explore the degree to which generalizable inferences can be drawn in the visual domain alone.

However, when inferences are drawn about the entirety of the visual diet or when the aim is to use existing datasets to build generalizable models of the visual system, including deep neural networks (DNNs), our results highlight clear constraints in existing datasets and caution against overinterpretation. This insight is particularly relevant given recent findings suggesting that DNN models of the visual system yield similar performance regardless of architectural differences or training objective (Conwell et al. (2024)), an insight that would strongly affect the neuroconnectionist research program that requires a system identification approach for identifying "better" models of the visual system (Doerig et al. (2023)). In contrast, our findings highlight that without sufficiently broad datasets, we cannot determine whether models truly converge or if their apparent similarity results from being evaluated on insufficiently diverse datasets.

How, then, can we design datasets that are generalizable? Our simulations indicate that, given a certain "stimulus budget", visual-semantic breadth of sampling should be prioritized over depth to ensure maximum possible OOD performance. Furthermore, all tested sampling strategies provided sufficient coverage of the stimulus pool, yielding comparable generalization performance. Notably, sampling strategy alone did not compensate for insufficient dataset diversity, emphasizing that future studies should prioritize broad stimulus pools even when using random sampling.

While we quantified diversity using CLIP embeddings, even the most diverse existing stimulus pools may omit crucial aspects of the visual representational space. Beyond CLIP, the field is in need of a more precise, quantitative definition of diversity to support broader, stratified sampling (Conwell et al. (2024)). It is also worth noting that many of our findings are based on simulations, where assumptions can affect results. While our results are consistent across simulations and validated with real fMRI data, future research should further empirically evaluate dataset diversity.

Overall, this study underscores the necessity of broader coverage in the representational space for making generalizable inferences than provided by existing datasets. By demonstrating that relatively small yet diverse stimulus sets provide large benefits for out-of-distribution generalization, we provide a framework for designing stimulus sets that enable large-scale, condition-rich studies of the visual-semantic system. Prioritizing diversity and coverage will allow researchers to construct datasets that better reflect the complexity of natural vision, leading to more robust models of how humans perceive the world.

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

## Code availability

All code used for the analyses and generation of figures in this study, including the pre-trained LAION-natural classifier mentioned in the text, is publicly available on GitHub: https://github.com/andropar/how-to-sample.

**Natural vs. Non-natural images**

A        Images considered natural             B        Images not considered natural

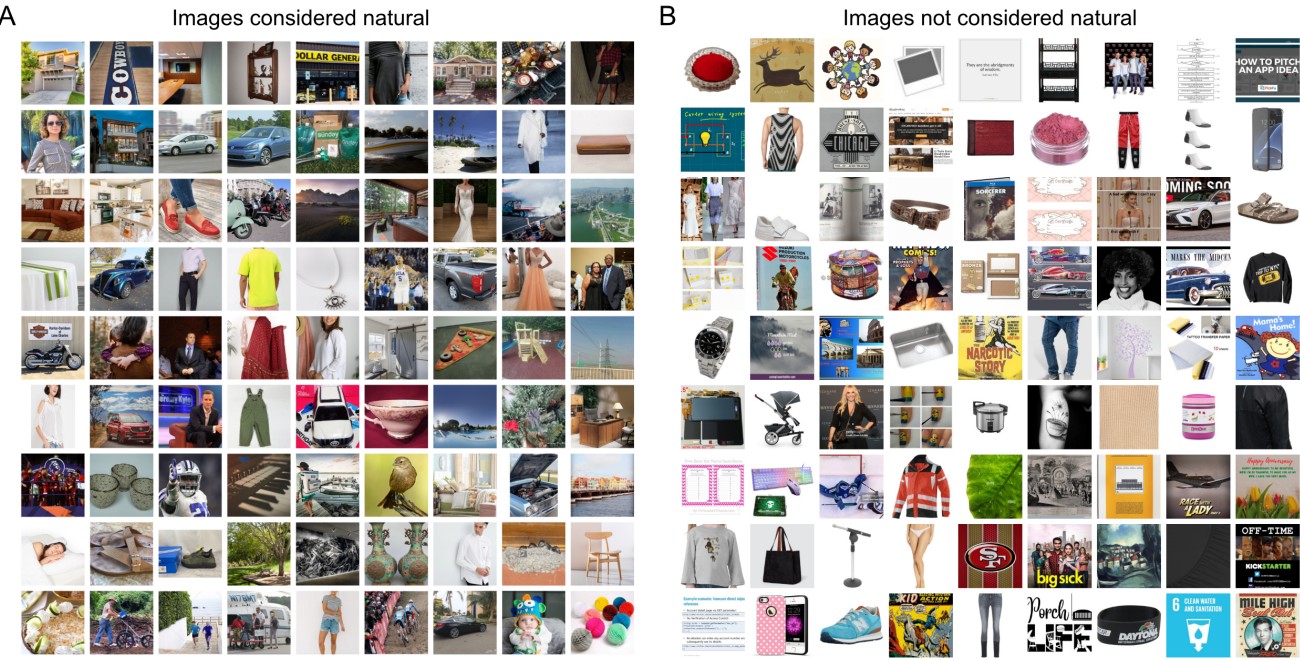

Figure S1: Examples of images from LAION-2B that were considered either natural (A) or not natural (B). The criteria for natural images were: "no heavy editing (e.g. high saturation / contrast, collages, cropped objects without background) or filter overlaid (e.g. black-and-white)", "no watermarks or text banners" and "must be a real-world object or scene (e.g. no screenshots of websites or video games)".

**Evaluation of OOD accuracy using NSD fMRI data**

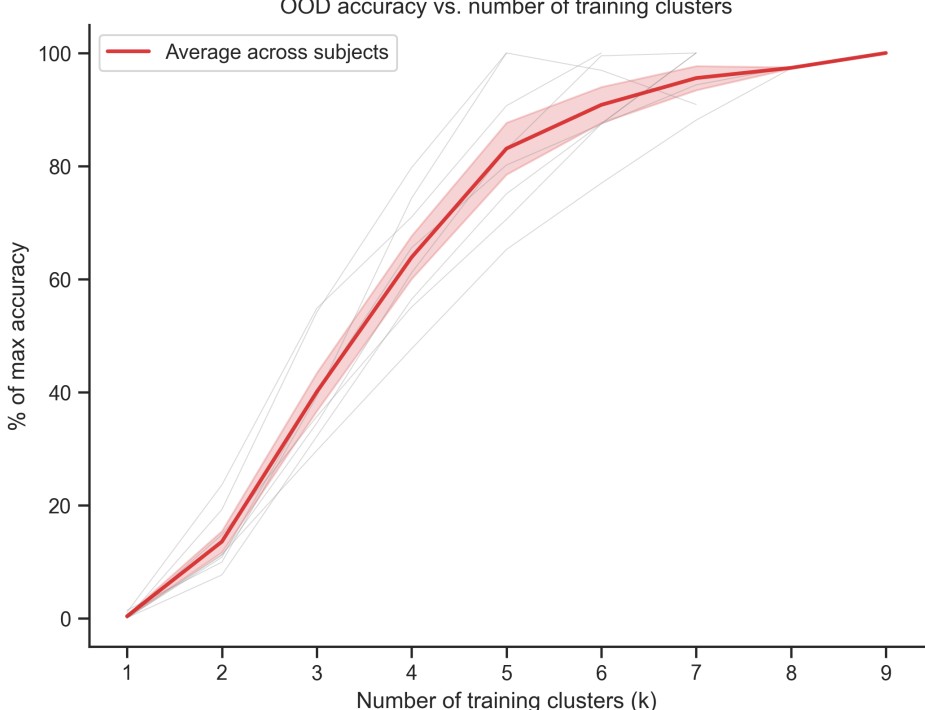

Figure S2: Impact of training set diversity on OOD accuracy in NSD. For each subject, CLIP features of presented images were clustered using mini-batch k-Means. Iteratively using one cluster as the OOD test set, training sets of a fixed size (N=500) were created by stratifying samples from a varying number (k) of the remaining clusters. Thin grey lines represent individual subject data, and the red line shows the mean across subjects, with shaded areas indicating the standard error of the mean. These results suggest that increased visual diversity improves generalization performance, even while keeping the total number of training samples constant.

**Examples of clusters not covered by THINGS or NSD**

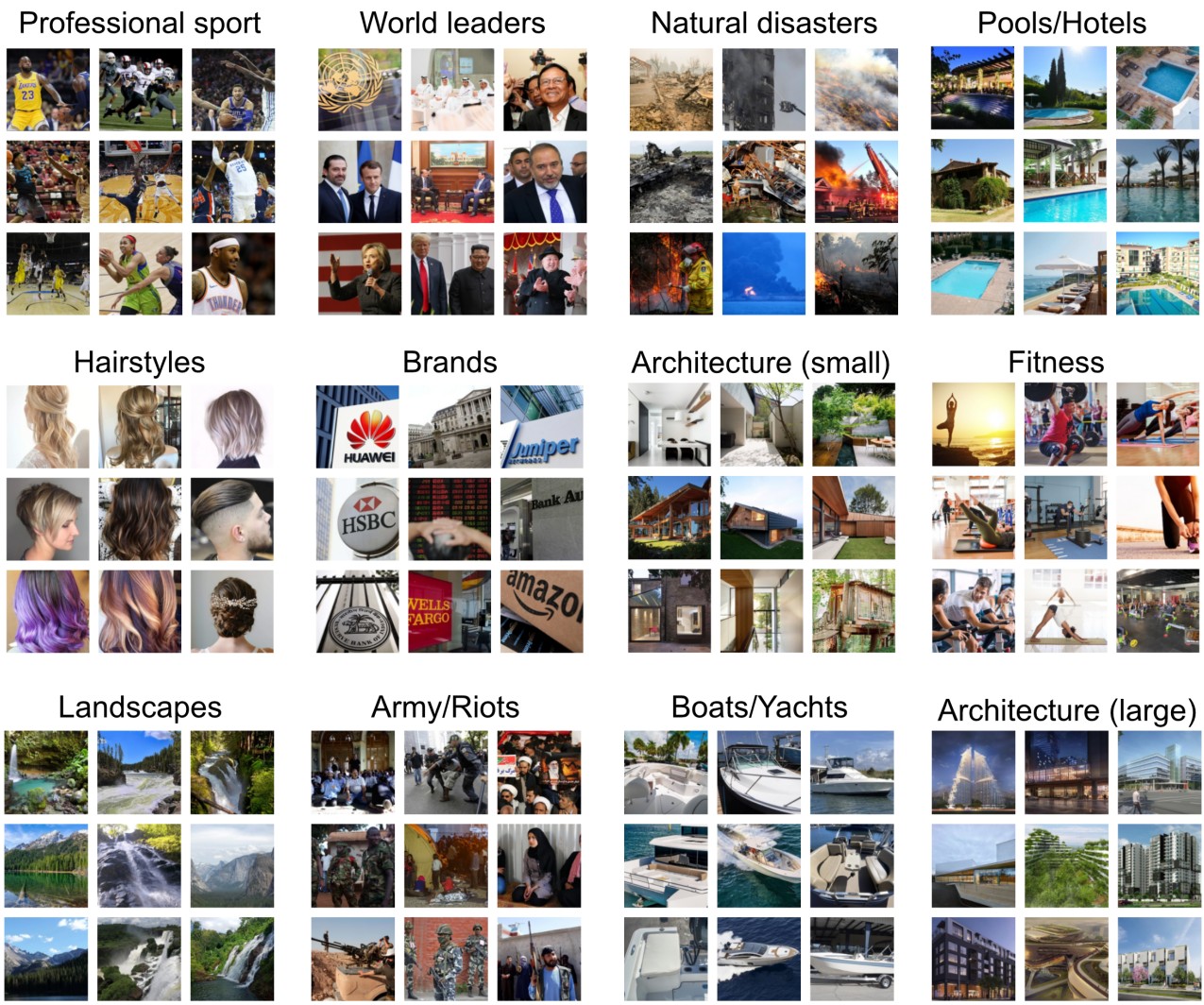

Figure S3: Examples of distinct clusters that were not covered by THINGS or NSD. These include certain sporting events, architectural styles, landscapes, political figures, images of natural disasters, activities and many more. Clusters were manually selected from the 50 largest clusters not covered by the other datasets, to avoid repetitions in semantic concepts.

