# OpenReview forum: "How to sample the world for understanding the visual system"
_ccneuro.org/CCN/2025/Proceedings — CCN 2025 Proceedings asProceedingsTalkPoster_

### Official Review · Reviewer_TnEv · 2025-03-28
**This manuscript addresses the crucial challenge of selecting appropriately diverse visual stimuli for robust generalization in visual-semantic neuroimaging studies, highlighting dataset size and diversity considerations through simulations and analyses. While the study makes important contributions, particularly emphasizing dataset diversity, methodological assumptions limit the interpretability and generalizability of the findings.**

**Soundness:** 2
**Clarity:** 2

**Comments:**

Interest:
The paper addresses an important and timely issue -- how to optimally select visual stimuli for fMRI studies to ensure broad generalization and semantic coverage. This question has wide relevance across computational cognitive neuroscience, particularly among researchers working at the intersection of vision, semantic representation, and modeling neural responses. However, given the specificity of using CLIP embeddings and image datasets, the paper might particularly appeal to those working directly with naturalistic imaging or visual-semantic models, rather than the broader neuroscience community. The contribution is significant in highlighting diversity and dataset size effects, though novelty is moderate as similar arguments about dataset generalizability have been made in the artificial neural networks literature for a longer time.

Soundness:
The claims made by the authors are generally supported by their extensive simulations. However, several critical assumptions potentially limit the robustness of their conclusions. Specifically, the reliance on CLIP embeddings as a proxy for the true conceptual space introduces a critical limitation: CLIP itself is trained on a corpus of images and captions with intrinsic biases, and thus reflects only the dimensionality spanned by its training data. Consequently, gaps identified by mapping NSD and THINGS onto LAION clusters could be artifacts arising from CLIP’s embedding space rather than genuine conceptual omissions. Importantly, the authors performed a one-directional mapping -- aligning NSD and THINGS onto LAION -- but not the reverse. Without mapping LAION embeddings back onto NSD and THINGS, it is unclear if these gaps represent genuine semantic omissions or merely differences in embedding density and distribution. This omission weakens the interpretability and generalizability of their core findings.

Additionally, the simplified assumptions in the simulations (e.g., conceptual space dimensionality of 512, 6 superclusters, uniform distribution of images within clusters, linear mapping plus Gaussian noise in synthetic fMRI responses) could significantly impact conclusions about practical dataset sizes. If any assumption diverges substantially from reality (e.g. conceptual features are not uniformly mapped onto the brain), the recommendation of ~6000 images as sufficient could be overly optimistic.

Clarity:
The paper is clearly structured, and most methods are described in sufficient detail for potential reproducibility. However, explicit discussion and justification for key methodological choices (e.g., dimensionality selection, directionality of mapping between datasets) would improve clarity and aid reproducibility. Given the complexity of the methods (especially simulations involving synthetic data), additional transparency or code availability would substantially enhance replicability.

**Expertise:**

2

**Interest:**

3

---

> ### Author Rebuttal · Authors · 2025-04-14
>
> We thank the reviewer for their helpful comments. Changes made to the manuscript are marked blue. Please note that this rebuttal had to be shortened a lot due to OpenReview character limits. We are happy to answer any remaining questions or concerns.
>
> > Interest: [...]
>
> We agree that our results could be skewed by CLIP biases. While domain-differences (NSD scenes, THINGS objects) appear in our coverage analyses, suggesting that uncovered clusters signify genuine representational gaps, we now also perform the reverse mapping.
>
> We clustered NSD/THINGS and tested coverage by LAION-natural. Using 10 million random LAION-natural points, 97.5% NSD/100% THINGS clusters were covered, with uncovered NSD clusters containing a single image. This supports LAION-natural covering NSD/THINGS and the significance of uncovered clusters from LAION-natural.
>
> > Additionally [...]
>
> First, we would like to highlight that we already provide a qualifying sentence in the discussion. While our analyses could not exhaustively test different parameters of the simulations we believe this does not diminish the validity of our findings, for the following reasons.
>
> The dimensionality of the conceptual space for the GMM-based simulation was set to the same dimensionality as the CLIP embeddings. We addressed uniform distribution of images within GMM-based clusters, by running the analysis on CLIP features from images from LAION-natural. We arbitrarily set the number of superclusters to 6, but cross-validated our analysis across all possible combinations of training clusters and OOD cluster. While we agree that a linear mapping plus Gaussian noise is a a strong assumption about the relationship between stimulus and brain activity, we would like to point out that this assumption is shared by most work in the field and is commonly made for encoding and decoding brain data.
>
> To further address these concerns, also in response to Reviewer #2, we added an additional result (Suppl., Fig. S3), which demonstrates that our key finding (“more diverse training sets lead to better OOD performance”) also replicates in real fMRI data. We hope that this provides further evidence for our findings.
>
> We also made changes to the manuscript to explain our choices and improve its clarity.
>
> >Clarity [...]
>
> We appreciate the suggestion to increase the transparency of our methods. We made adjustments to the paper to better explain our choices, and we will also make the analysis code available upon publication.

---

> > ### Comment · Reviewer_TnEv · 2025-04-18
> >
> > Thank you for the revisions and the rebuttal. I find the submission more convincing now, especially with the addition of the reverse analysis - mapping LAION embeddings back onto NSD and THINGS - which demonstrates the high coverage of these datasets by LAION. The analysis of OOD accuracy using NSD fMRI data also strengthens the argument that a large number of clusters in the dataset promotes better generalization.

---

> > > ### Author Response · Authors · 2025-04-21
> > >
> > > Thanks a lot for the positive review and the positive rebuttal comment regarding to soundness and clarity of our work!

---

### Official Review · Reviewer_PXqU · 2025-03-29
**Clarity and Completeness Concerns**

**Soundness:** 2
**Clarity:** 2

**Comments:**

This work presents a novel data (sub) set and methodology for the selection of natural images from a larger dataset (LAION-natural from LAION-2B), and results on the requirement for dataset diversity and sufficient number of samples to achieve OOD generalisation.

Overall this paper touches on some important and interesting topics regarding diversity and coverage of datasets (especially in the context of conclusions drawn from neuro-imaging datasets). However, a number of aspects of this paper lack clarity (which can be improved by modifying the text) and a number of conclusions appear to be unsupported or lack complete testing and soundness (to be improved by expanding results).

**Clarity Concerns**

First, in terms of clarity there are a number of things which could be improved. Most prominently, the paper begins by laying out out it’s contributions, claiming in the abstract that they “… reveals significant representational gaps in existing datasets …”, “… show that these gaps lead to impaired out-of-distribution generalisation”, and that this work will “… provide clear strategies for future studies …”. Given the results and some lack of broader comparisons, these claims seem to be overblown.

In particular, the impairment on OOD has not been shown in real-world tasks but in a toy-model, and the conclusions of this study do not appear to be entirely clear in strategy which one should use. Simply tempering these claims early in the work (Title as well as Abstract and Introduction) and laying out this as more of an exploration that provides interesting hints toward better approaches, would be more appropriate for its true contribution. The discussion section already does this rather well by cautioning a reader on the conclusions drawn from simulated figures, for example.

Next, the conclusions which one should take from this work are rather vague. First, indeed a sufficient number of samples is important - however this work suggests that between 5-10k samples would be necessary for good OOD generalisation. Within experimental conditions this would be extremely challenging to collect data for. Furthermore, the conclusions on the importance of diversity is clear but how to measure or conclude ‘sufficient diversity’ is not so clear. I also cover a contradiction on the dataset diversity in my concerns on soundness below.

Furthermore, a number of things are unclear in how they are written or presented. First, the distinction of what counts as ‘natural’ in this work is unclear unless one reads the methods - this could be better explicated and defended early in the paper (Introduction). On this note, is even unclear whether the category NSFW should be considered natural or not and this is never discussed (though sensitivity of this category should naturally be considered). Second, a number of aspects of figures are unclear: at some point the term ‘LAION’ is used in figures (1, 2 and 3) to exclusively refer to LAION-natural but this is rather unclear since LAION-2B also exists. Finally, Figure 2F is not provided an explanation in the caption and, “ED” is quoted in Figure 3’s legends but without any explanation in the results text or caption (only in methods).


**Soundness Concerns**

Figure 2 appears to show that a random sampling of images from LAION-natural allows for as much OOD generalisation than a stratified sampling. Figure 3 expands on this and shows that a wide range of sampling strategies are sufficient. However, this raises an important question for this work: if the diversity of the underlying dataset is most important (and not the sampling strategy), is the inclusion of non-natural as well as natural images superior to natural images alone? It seems that it would be possible to test such a conclusion in Figure 2 quite simply by comparison against LAION-2B. In all, it appears that the results did not consider testing whether the properties of LAION-natural are superior (or not) to a random subset of the samples from LAION-2B - an important consideration since the strength of dataset diversity is the final and most coherent conclusion.

For an example of the utility of such a comparison, see how EcoSet (Mehrer et al. 2021) compared against the full ImageNet dataset from which they sourced their data.


Overall this is an interesting area of research and some conclusions here (as well as the dataset construction) are a nice contribution. However, there is also a lack of completeness as yet by my estimation.

**Expertise:**

2

**Interest:**

3

---

> ### Author Rebuttal · Authors · 2025-04-14
>
> We thank the reviewer for their helpful comments. Changes made to the manuscript are marked blue. Please note that this rebuttal had to be shortened a lot due to OpenReview character limits. We are happy to answer any remaining questions or concerns.
>
> > First, in terms of clarity [...]
>
> We agree in part and toned down the abstract, but believe our claims are supported by our results. Based on the reviewer’s comment, we have added further evidence or clarification.
>
> The OOD generalization conclusion follows from thorough simulations (cf. Shirakawa et al., 2024), showing that less diversity negatively impacts OOD generalisation. We agree that we did not show this on real fMRI data. To address this, we added an additional analysis (Suppl., Fig. S3), replicating simulation findings in fMRI data from NSD.
>
> We agree that we did not specify how to operationalize on our proposed strategy (diversifying the stimulus set as much as possible) and have softened the language regarding this point.
>
> > Next, the conclusions which [...]
>
> We agree that 5-10k samples is a large number of conditions, but want to point out that our guidance targets such large-scale studies. We also agree that we don't define “sufficient diversity”. Our point, now clarified in the Discussion, is that given a certain “stimulus budget”, it should be distributed across as much of the visual-semantic space as possible.
>
> > Furthermore, a number of things [...]
>
> We changed the paper to better explain the filtering process to obtain LAION-natural and added missing details to the figures.
>
> > Figure 2 appears [...]
>
> Thank you for raising this important question. LAION-natural was created in a two-stage filtering process: removing any image that is not suitable for vision studies and removing any image that can’t be considered a photograph. Our aim was to create a proxy of natural visual experience, for the purpose of comparing existing naturalistic fMRI datasets against it. We agree that, given our findings, these filtered-out images could be valuable additions for a stimulus set that can also contain non-naturalistic stimuli (& NSFW images too) - however, this is not guaranteed, as the content of all non-photograph images could appear as part of a photograph.
>
> To test this, we computed the coverage of LAION-2B by random subsets of LAION-natural and found an average coverage of 93.14%, slightly smaller than that of LAION-natural. This suggests that most of the variance in LAION-2B is captured by it.

---

> > ### Comment · Reviewer_PXqU · 2025-04-18
> >
> > _(This is TPC providing a pointer to the [**Official Comment**](https://openreview.net/forum?id=T9k6KkZoca&noteId=EcmCfkw3Zn) from Reviewer PXqU starting with:_
> >
> > > Thank you for your rebuttal. I think ...
> >
> > _With this **Rebuttal Comment** posted on behalf of the Reviewer, the Authors can respond with one final **Reply Rebuttal Comment** during the Author-Reviewer Discussion phase.)_

---

> > > ### Author Response · Authors · 2025-04-18
> > >
> > > Thank you for the follow-up and clarifying your concern about the necessity of filtering LAION-2B to naturalistic images. We agree that LAION-2B could be more diverse than LAION-natural. To test for this we provide the same metric as in Fig. 1D for LAION-2B in the revised manuscript:
> > >
> > > > [...] we computed the coverage of LAION-2B by random subsets of LAION-natural and found an average coverage of 93.14%, slightly smaller than that of LAION-natural. This suggests that most of the variance in LAION-2B is captured by it.
> > >
> > > We incorporated this finding into the paper in “Results - Significant gaps in visual-semantic coverage in existing stimulus sets.” and believe it provides sufficient evidence for LAION-natural not removing significant parts of the representational space of LAION-2B. However, as suggested by you, we now also ran the analysis from Fig. 2E again, measuring the OOD performance of random NSD-sized subsets of LAION-2B. We found that LAION-2B outperforms LAION-natural by only ~2.34%, in line with the previous analysis. If possible, we will add this analysis to the supplemental materials of the paper for the camera-ready version.
> > >
> > > Additionally, we would like to point out that we are not arguing for LAION-natural being the best, or most diverse, stimulus pool, but use it as a reference dataset to compare other datasets against. As THINGS and NSD only include photographs, naturalistic filtering enabled a fair comparison within this domain. Since LAION-natural encompasses THINGS and NSD, this choice also does not affect our main conclusions.
> > >
> > > However, we appreciate the thought that filtering was not required to achieve the two properties you mentioned. As LAION-2B is a superset of LAION-natural, this should be true by default. For a stimulus set that is not limited to naturalistic images, using non-naturalistic images from LAION-2B might be beneficial, given our findings.
> > >
> > > We hope this clarifies our reasoning and addresses your remaining concern. Thank you again for your feedback which has helped strengthen the paper.

---

### Official Review · Reviewer_jN3M · 2025-04-01
**Interesting analyses of an important new dataset!**

**Soundness:** 2
**Clarity:** 3

**Comments:**

In an interesting paper, the authors introduce the database "LAION-natural", which is a 120-million-image subset of the larger LAION-2B database. The motivation behind creating this dataset was to achieve greater visual-semantic diversity, ensuring that any inferences drawn based on it would generalize. In addition to introducing and describing the dataset, the authors conduct several key analyses. Among others, they examine how well previous datasets cover the diversity of concepts contained in LAION-natural, and they also assess the merit of the dataset when sub-sampled to a scale that is more realistic for human experimentation.

I believe this is a strong paper that should be of interest to many CCN attendees, whether they focus on deep learning, human experiments, or methods of stimulus set creation. I find the dataset, the analyses, and the conclusions to be significant. The dataset creation is rigorous and systematic, as are the analyses. Moreover, the paper is generally well-written. I have only a few comments regarding some of these analyses and certain aspects of clarity in writing:

1. A main result of the first analysis is that THINGS and NSD show large gaps in how they cover the visual-semantic space defined by LAION-natural (with the slightly higher coverage of NSD possibly attributable to its larger dataset size). This finding is not very surprising to me, given the relatively small sizes of those two databases. I appreciate the sub-sampling in later analyses (and, for instance, inclusion of Fig 2E), but I do wonder about the influence of dataset size and certain analysis parameters in the context of Figure 1 directly. For example, is using 5500 clusters appropriate given the small size of THINGS (and how would results be for a smaller cluster set size)? Similarly, I also wonder how a random LAION-natural subsample of only ~70k or ~26k images would behave in terms of coverage of the visual-semantic space defined by the full LAION-natural dataset, as quantified in Figure 1C. (I understand that running these analyses may be too complex given the CCN review timeline, but I feel that even just commenting on this issue would benefit the paper.)

2. In light of these considerations, I also think some of the language around these results could be slightly adjusted. I liked the statement in the discussion ("However, this does not diminish the value of these datasets (...)") and believe it would help to highlight this perspective already earlier on. For instance, it is not necessarily the goal of THINGS/NSD to cover the entire semantic space; they may be aiming for a more constrained, systematically validated set in other respects.

3. It is mentioned that all concepts from ImageNet, MS COCO, and Ecoset were incorporated. Given, for instance, the numerous dog subclasses in ImageNet, I was a bit surprised about this decision (assuming I understood correctly that this would mean that all dog types, rather than just the big category "dog", are included as concepts), especially considering that Ecoset specifically seeks to include only higher-level categories. While this may pose no issue for a very large dataset of millions of images, I wonder how it might impact the coverage of the visual-semantic space when significantly sub-sampling. Of course, I am not suggesting changes to the dataset itself but thought that elaboration on this could potentially be helpful.

4. The paper mentions that some of the concepts not covered by the other databases include landscapes, natural events, crowds, or non-Western public figures. I find this to be a very interesting result and would love to learn more about the details of these gaps. Perhaps, they could be offered in supplementary material or on a future website for the database, as it would provide interesting insights into what is missing from existing datasets and what LAION-natural adds.

Overall, I believe this is an important contribution and I recommend it for publication.



### After author response and other revisions:
Increased clarity to "exceptional".

**Expertise:**

2

**Interest:**

3

---

> ### Author Rebuttal · Authors · 2025-04-14
>
> Thank you for the positive feedback and these questions. To address the comments:
>
> 1) We would like to point out that the analysis in Figure 1C is based on a very small subset of LAION-natural (6k samples), which already achieves a coverage of >90% of the full dataset. In comparison, THINGS and NSD (full-sized) only achieve a fraction of the coverage. This suggests that the small size of the datasets itself is not the reason for the low coverage of LAION-natural by these datasets, but instead a lack of certain visual-semantic concepts. We now revised the manuscript to better highlight this distinction and the specifics of the analysis.
>
> 2) Thank you for highlighting the contributions of THINGS/NSD and correctly pointing out that their goal is not necessarily a full coverage of the semantic space. We agree that these datasets are already extremely useful for many types of analyses. However, they are frequently used to make generalizing statements (for example about the reconstructability of stimuli from brain activity) and, in line with prior work, we show that they may have limitations that researchers may want to be aware of in this regard. We now revised the manuscript to better highlight the value of THINGS/NSD in the context of our contribution.
>
> 3) Thank you for raising this issue. Please note that we did not incorporate all concepts from these datasets, but instead simply tested if they occur in LAION-natural at all. Our aim here was to ensure that LAION-natural contains all semantic concepts found in these other datasets per default, no matter their granularity. We have now adapted the manuscript to clarify this aspect of our work.
>
> 4) Thank you for this suggestion, we agree that adding more details on the representational gaps is a great idea. We have now added an overview of 12 large and distinct clusters that are not covered by THINGS or NSD to the Supplementary Material (S4), including quite basic concepts like architecture or landscapes, but also more specific concepts like brands, fitness, or hairstyles.
>
> We believe these comments and questions greatly helped us with improving the clarity of our paper. Any changes we made to the revised manuscript are highlighted in blue.

---

> > ### Comment · Reviewer_jN3M · 2025-04-18
> >
> > _(This is TPC providing a copy of the [**Official Comment**](https://openreview.net/forum?id=T9k6KkZoca&noteId=0rFNqQjw45) from Reviewer jN3M with:_
> >
> > > I would like to thank the authors for their excellent rebuttal (my apologies also for the oversight regarding point 1). I am very happy with the revised version of the paper!
> >
> > _With this **Rebuttal Comment** posted on behalf of the Reviewer, the Authors can respond with one final **Reply Rebuttal Comment** during the Author-Reviewer Discussion phase.)_

---

> > > ### Author Response · Authors · 2025-04-21
> > >
> > > Thanks again for the positive review and the very positive response to our rebuttal regarding the soundness of our work!

---

### Meta-Review · Area_Chair_duAQ · 2025-05-05

**Ccn Recommendation:** Accept as Proceedings

**Metareview:**

All reviewers found that this would be of broad interest to the CCN audience. Quantifying the overlap between existing datasets seems particularly valuable to the community. Although the paper was a bit borderline for the CCN-P track this year, we have decided to **accept** the submission for presentation. Congratulations and we look forward to seeing you at the conference!

One note for final revisions: while the analysis in Figure S3 analysis was interesting to compare with real-brain data, as an AC I found that it lacked sufficient detail and was not clear (for instance, OOD cluster accuracy on the y-axis goes to 0.08, but from the caption it seems like there are only 10 clusters in the test data, so this is still below chance 1/10?). This analysis should be revised and clarified in a final version. Please also include the stated comparison against LAION-2B (mentioned in the response to PXqU) in the supplement, as stated in your response.

**Summary:**

This submission subsamples the LAION-2B image dataset into a set of 120 million images that the authors introduce as “LAION-natural.” Using this dataset, the authors analyze how increasing the dataset diversity can lead to out-of-distribution generalization. Reviewers thought this would be of broad interest to the CCN audience.

All reviewers provided suggested changes for the authors to increase the clarity of the paper, and the authors added some additional methods details and discussion to the text. There were also concerns about the soundness of the results, and the authors added additional analysis (1) reporting a reverse mapping from LAION embeddings back to NSD and THINGS and (2) reporting potential differences in coverage between LAION-2B and LAION-natural. Reviewers also pointed out that the conclusions of the study are based on simulations with simplified assumptions, which may not be sufficient for real brain-data collection. The authors presented an analysis (Figure S3) in an attempt to address this concern, and also added some additional text to the discussion. Reviewers generally found that the author rebuttals addressed their concerns, and both reviewers and authors were active in the discussion period.

**Expertise:**

3